



# ESD Ideas: Global climate response scenarios for IPCC AR6

Rowan T Sutton and Ed Hawkins

National Centre for Atmospheric Science, Department of Meteorology, University of Reading, Reading, RG6 6BB. UK

*Correspondence to*: Rowan Sutton (rowan.sutton@ncas.ac.uk)

**Abstract.** Policy making on climate change routinely employs socio-economic scenarios to sample the uncertainty in future forcing of the climate system, but the IPCC has not developed similar discrete scenarios to sample the uncertainty in the global climate response. Here we argue that to enable development of robust policies this gap should be addressed, and we propose a simple methodology.

The Working Group I (WGI) contribution to the 6[th] Assessment Report (AR6) of the Intergovernmental Panel on Climate Change (IPCC) is in preparation for publication in April 2021. One of the requirements is to provide *assessed* projections of global climate. Such projections depend on future forcing of the climate system and on the response to this forcing (Hawkins and Sutton, 2009). Traditionally, uncertainty in future forcing is sampled using a discrete set of socioeconomic scenarios, whereas uncertainty in the climate response is sampled using multi-model projections from the WCRP Coupled Model Inter-

comparison Project (CMIP). The presentation of "raw" (i.e. uncorrected) CMIP projections has been supported in several previous WGI reports by an *assessment* that the 5-95% CMIP range is the *likely* range (66% probability) of the future climate response, at least for the long term. However, emerging results from CMIP6 suggest that a similar assessment is unlikely to be tenable for AR6. (In particular, several models show significantly higher Equilibrium Climate Sensitivity (ECS) than the previous generation of CMIP5 models, and their ECS values fall outside the AR5 assessed *likely* range (Forster et al, 2019)).

Furthermore, a primary focus on the *likely* range for future climate is in any case ill-suited to the needs of policy makers faced by problems of risk assessment (Sutton, 2018; 2019). In risk assessment there is special interest in high impact scenarios, even if their likelihood is considered low (King et al, 2015).

To address the needs of risk assessment, Sutton (2019) proposed that IPCC WGI should develop a discrete set of scenarios to

sample uncertainty in the global climate response, analogous to the socio-economic scenarios used to sample forcing uncertainty. This idea can also address the challenge for AR6 to present global climate projections that are consistent with the assessment of key parameters such as ECS.[1] Here we present a simple demonstration of how this could be done for projections of global mean surface air temperature (GSAT), exploiting the CMIP6 projections and estimates of ECS for each model.

---

[1] A further attraction of basing global response scenarios on ECS is that some of the same scenarios could be used in multiple assessment cycles, providing policy makers with helpful continuity between reports.





For each of the socio-economic scenarios (SSPs), we regress the simulated mean GSAT change onto ECS from each CMIP6 model in each overlapping 20-year period (with central years 2025-2090, examples in Fig 1 a, b). The slope of this regression defines the climate response scenario for each SSP and time period (panel c) and can be used to produce GSAT projections as a function of ECS (climate response scenario) and SSP (emissions scenario). Fig 1c shows projections and climate response scenarios for GSAT change under SSP1-2.6 and SSP5-8.5. In each case, the 5-95% range spanned by the currently available

CMIP6 models is shaded, and three response scenarios are shown corresponding to ECS values of 2 °C, 4 °C and 5 °C. No quantitative likelihood is attached to each scenario and there is no "best estimate" – they are merely chosen to illustrate a range of possibilities relevant to risk assessment. For the purposes of discussion, we imagine that the AR6 assessment is that ECS is *likely* in the range 2.5 to 4.0°C, and *very likely* in the range 2.0 to 5.0°C. Thus the 4 °C ECS scenario corresponds to the upper end of our *likely* range. As impacts and risks have been assessed to increase rapidly with GSAT (e.g. the "burning

embers" figure - IPCC, 2014), it could be used to estimate the highest impacts consistent with the assessed likely range. The 5 °C ECS scenario may be considered a Physically Plausible High Impact Scenario, in line with the definition of Sutton (2018). It corresponds to a highly sensitive climate system leading to rapid warming and rapidly increasing risks and associated costs of adaptation and/or mitigation. Under the 2°C ECS scenario, the direct impacts and costs of climate change would be less severe, or delayed. However, it might still be considered high impact from a policy point of view as it could imply that the

costs of adaptation and mitigation would be lower than previously anticipated.

Fig 1 also illustrates projections for the most and least rapidly warming models under each SSPs. In the absence of counter evidence, these projections might also be considered physically plausible, so these projections offer alternative - more extreme - choices for high impact scenarios. However, such scenarios are likely to be less robust because of their reliance on single

model results.

To inform risk assessments, scenarios must be combined with quantification of impacts. There is no single metric of impact: many, many, variables are relevant to policy and decision making. As a simple illustration, we consider here the time of crossing specific temperature thresholds. This variable is particularly important for climate policy following the framing of

the Paris Agreement in terms of ambitions to stay below specific levels of GSAT relative to pre-industrial climate. Fig 1d,e illustrate the year in which the 2°C and 3°C warming thresholds are crossed, as a function of socio-economic scenario and climate response scenario. It is immediately apparent that whether and when the thresholds are crossed depends as much on the response scenario as on the forcing scenario. For example: under SSP 2.6, the 2.0°C threshold is only crossed under the highest ECS scenarios; under SSP 7.0 and 8.5 the 5°C ECS scenario yields crossing times 2-3 decades earlier than the 2°C

ECS scenario. A notable feature of panel c is that, before 2060, the 5°C ECS scenario for SSP 2.6 (low emissions) is warmer than the 2°C ECS scenario for SSP 8.5 (high emissions). All these results illustrate very clearly that ***climate response scenarios are just as relevant to mitigation policy as are socio-economic scenarios***. The development of robust policies must consider both factors, including explicit attention to high impact scenarios, such as the 2°C and 5°C ECS scenarios considered here.



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

**Acknowledgements**
We thank Christine McKenna and the CONSTRAIN project for providing CMIP6 GSAT CMIP6 data and Kari Alterskjar and Piers Forster for providing the CMIP6 ECS values.

**Fig 1 Scenarios for GSAT derived from CMIP6 projections.** Panels a, b show regressions, for each SSP, of simulated mean GSAT for a range of models onto each model's own estimated ECS value, for two example 20-year periods. Where multiple
ensemble members are available we have used the ensemble mean response. The simulations are first referenced to the mean of 1995-2014, and baselined to an approximate pre-industrial level using an observed change from 1850-1900 to 1995-2014 of 0.76°C using HadCRUT4 (Morice et al. 2012). Panel c shows three climate response scenarios (thick lines) assuming ECS values of 2, 4 and 5C, for two SSPs (SSP5-8.5, red, and SSP1-2.6, blue), along with the 5-95% simulated range (shaded) and the simulations with the largest and smallest responses at the end of the century (thin lines). Panels d, e show the decade in
which 2C and 3C GSAT thresholds are first crossed as a function of climate response scenario and emissions scenario. Grey shading indicates that the threshold is not crossed by 2090 (i.e. the 20-year average of 2081-2100). Note that: (1) the SSPs are not equally spaced in terms of radiative forcing, (2) GSAT declines in the latter part of the century for some SSPs and, in some cases, may fall back below one of the thresholds shown, but we do not include that possibility in panels d, e, and (3) a different reference period choice would produce different ranges, especially for the near-term; we do not consider this sensitivity here
and do not analyse a threshold crossing of 1.5°C for this reason.



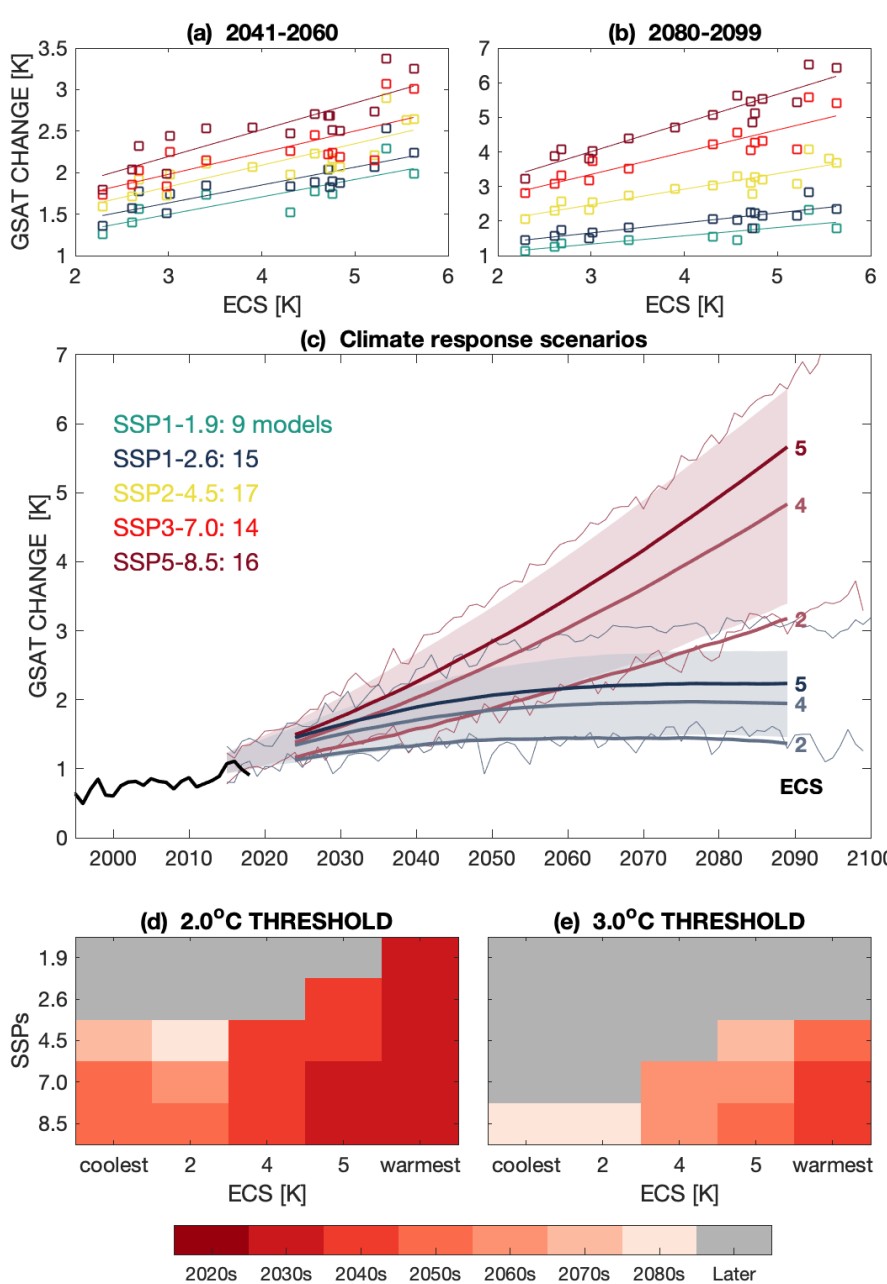

**Figure 1: Scenarios for GSAT derived from CMIP6 projections**