# Peer review of "ESD Ideas: Global climate response scenarios for IPCC AR6"

_Earth System Dynamics, 2019_

## Referee Comment (RC1) · Anonymous Referee #1 · 4 Mar 2020

Review Comments on "ESD Ideas: Global climate response scenarios for IPCC AR6"

This article proposes a new methodology to fill the gap of the current existing IPCC scenarios. I should admit that I am not a climate modeler and have expertise in integrated assessment modeling. Thus I can comment from that perspective and need other climate expert's points of view. I list the main arguments below.

1) The current title and abstract lead misunderstanding of the contents and should be reconsidered. In the abstract, there is a problem statement "but the IPCC has not developed similar discrete scenarios". However, the issue is wrongly stated. First, after SRES IPCC never develops scenarios. IAMC (Integrated Assessment Modeling Consortium) generated RCPs and SSPs during the last decade but they are certainly different from IPCC. Second, assuming that the RCPs and SSPs are a set of scenarios

[Figure]

addressed by the authors here, the authors are correct within the CMIP context in a way that the socioeconomic scenarios are not discrete because the full-scale climate models cannot be run for such a large number of scenarios. To this end, a series of past IPCC assessment reports used simple climate models to represent CMIP3 and CMIP5 climate model behavior with ranges of embedded parameter ensembles. So, the issue is neither nonexistence of discrete scenarios nor how to use the scenarios. I think what the authors demonstrate in this paper is just one of the examples of how to use the CMIP results (not the scenario issue). 2) Following the above point, the series of IPCC reports in WG3 has shown the climate implications with probability and uncertain ranges of climate models. In the assessment, climate sensitivity has been already considered to generate the parameter ensemble of simple climate models sampling various parameters simultaneously. Thus, figure 1 panel c has been already addressed. The new thing here would be to show the climate outcomes explicitly comparing with climate sensitivity. Perhaps, it would be new but it needs confirmation from the climate expert. 3) Although it is not the IPCC coordinated activity, it would be worthwhile to acknowledge that in integrated assessment modeling, the discrete scenario proposal has been made and they are now working on that1. 4) Finally, in the paper, I can find the term "each SSPs", but it seems to be a set of SSPs and climate target combination by looking at the figure. For example, SSP1-26 is a combination of SSP1 and radiative forcing target 2.6W.

1. Fujimori S, Rogelj J, Krey V, Riahi K. A new generation of emissions scenarios should cover blind spots in the carbon budget space. Nature climate change 2019, 9(11): 798-800.

---

## Referee Comment (RC2) · Anonymous Referee #2 · 7 Apr 2020

Review ESD Ideas: Global climate response scenarios for IPCC AR6 by Rowan Sutton and Ed Hawkins.

This article builds on earlier articles (Sutton, 2018, 2019) that emphasize the need for climate risk assessments by the climate modelling community and specifically by WGI of IPCC. In Sutton 2019 specific priorities for WGI are outlined and one of those is to "Develop a discrete set of global climate scenarios". This article provides a methodology to generate such a discrete set of scenarios based on ECS.

To my opinion this is a very valuable contribution to this discussion that is timely needed. The proposed method can be easily applied not only to GSAT but also to other variables. The article is well written and the ideas are clearly illustrated and elaborated. The presentation of the ideas in the figure is powerful and will stimulate the

scientific debate.

There is, however, one aspect that to my opinion deserves a bit more discussion. Climate change is already happening. This means that climate models can be evaluated against their ability to simulate historical trends. How should models be treated that deviate substantially from the historical trend (e.g. high ECS models)? I realize that this is not an easy topic that involves issues as natural variability, compensating errors and warming mechanisms that may become relevant at future warming levels. Some discussion is to my opinion, however, needed because the historical warming is one of the few observational checks of the simulated greenhouse warming by the models. Scenarios do. by their nature, not involve likelyhood, but if models are unable to represent the observed past this creates a tension with the "plausibility" of the scenarios.

Typo: Line 69: Reference Hawkins and Sutton, should be 2009 and not 2019

---

## Referee Comment (RC3) · Anonymous Referee #3 · 6 May 2020

The authors present a useful approach to explore how climate impacts jointly depend on forcing scenario (as encapsulated in SSP socio-economic pathways) and on climate response scenario (as encapsulated in equilibrium climate sensitivity as determined using abrupt 4xCO2 experiments). This could be published as-is, but I offer the following minor comments for the authors to consider.

The authors might comment on the non-monotonic behavior exhibited in Figure 1d. Why is there a local maximum in crossing time for ECS=2K models (i.e., crossing the 2ËŽC threshold sooner if ECS is lower or higher than 2K)? Is this just due to the limited sample size of models that performed all of the experiments necessary to make the figure (multiple SSPs and abrupt-4xCO2). Do the results change materially if the analysis is restricted to only the 9 models that are present in all experiments (it appears

that the limiting experiment is SSP1-1.9). Perhaps as more models get published, this noise will beat down? I suppose that the smaller the warming threshold, the greater the possibility that internal variability or inter-model variability could give rise to such non-monotonic behavior when there is such a limited model sample size.

Suggest labeling the ECS bin edges instead of the midpoints, as they are not evenly spaced and it is unclear what marks the transitions, especially from 5K to "warmest" and from "coolest" to 2K.

Would there be any value in making a version of panels (d) and (e), but showing the GSAT change relative to the baseline at 2050 and at 2100, as a joint function of ECS and SSP? This is basically what is shown in panels (a) and (b), but might be clearer if presented like (d) and (e).

While I understand the motivation and appeal of exploring the joint dependence of a climate impact on two "scenarios" – socio-economic and climate response (or forcing and response), I'm not really a fan of the phrase "climate response scenario" and would prefer that "scenario" be reserved for SSPs. Scenarios in the climate context have historically referred to plausible future social-economic futures (SRES, RCP, SSP) for which humans have some role in determining. ECS is a different beast. Could it just be referred to as "climate response"?

---

## Referee Comment (RC4) · Anonymous Referee #4 · 11 May 2020

This paper by Sutton and Hawkins presents a new proposal to better reflect geophysical uncertainties that surround the socioeconomic scenario uncertainty in projections of future climate change. The aim of the proposal is to provide more useful information for risk assessments.

The paper in my view can potentially provide a useful contribution to the framing of scenario projections. The suggested visualisation that shows how different thresholds are exceeded as a function of both the forcing and the uncertainty in the response, can provide useful additional insights to users of climate change data. At the same time, the contribution is very succinct and important aspects that are key components of risk assessments are not being covered or discussed. An important aspect in risk assessments is the likelihood connected to a certain hazard. In the proposal presented here,

this dimension is not reflected upon. The authors have selected five different ECS levels, and represent them with equal weight. The same is true for the scenarios for which it is obvious that not all are equally likely. This raises immediate additional questions and issues. In particular, the suggested approach creates a great opportunity for biased representation of scientific evidence, which is skewed towards arbitrary selected extremes.

It would therefore be useful if the authors could give this aspect a bit more thought and come up with a suggestion of how likelihood (at least in the geophysical sense) could be incorporated in the visualisations in support of risk assessments.

---

## Author Comment (AC1) · 18 May 2020

We thank the referee for their thoughtful comments on our article. We repeat the comments here, together with our responses.

—

This article proposes a new methodology to fill the gap of the current existing IPCC scenarios. I should admit that I am not a climate modeler and have expertise in integrated assessment modeling. Thus I can comment from that perspective and need other climate expert's points of view. I list the main arguments below.

1) The current title and abstract lead misunderstanding of the contents and should be reconsidered. In the abstract, there is a problem statement "but the IPCC has not developed similar discrete scenarios". However, the issue is wrongly stated. First,after SRES IPCC never develops scenarios. IAMC (Integrated Assessment Modeling Consortium) generated RCPs and SSPs during the last decade but they are certainly different from IPCC. Second, assuming that the RCPs and SSPs are a set of scenarios addressed by the authors here, the authors are correct within the CMIP context in a way that the socioeconomic scenarios are not discrete because the full-scale climate models cannot be run for such a large number of scenarios. To this end, a series of past IPCC assessment reports used simple climate models to represent CMIP3 and CMIP5 climate model behavior with ranges of embedded parameter ensembles. So, the issue is neither nonexistence of discrete scenarios nor how to use the scenarios. I think what the authors demonstrate in this paper is just one of the examples of how to use the CMIP results (not the scenario issue).

RESPONSE: We accept that IPCC did not itself generate the RCPs and SSPs and will correct this point. We also agree that simpler climate models have often been used to explore a wider range of socio-economic scenarios than is possible with complex (CMIP-class) climate models, although WGI (which is the focus of our Idea) has placed great emphasis on the specific set of socio-economic scenarios used for CMIP.

However, the focus of our Idea is not socio-economic scenarios at all but rather the representation by IPCC of information about uncertainty in the climate response to anthropogenic forcing. In past assessment reports this information has usually been expressed in terms of a likely range based on CMIP results, e.g. the likely range for global mean surface temperatures under a particular socio-economic scenario. Our argument (following Sutton 2018 & 2019) is that such an approach is flawed because it does not meet policy-maker needs for risk assessments which require specific attention to high impact scenarios even if they are considered unlikely to arise (noting that according to IPCC calibrated language unlikely means only <=33% chance). We therefore propose as an alternative that IPCC should develop and exploit a set of scenarios to sample the uncertainty in the climate response, and in doing so should pay specific

attention to high impact scenarios. Our paper then explains a simple way in which this can be done.

We have modified the abstract and introductory paragraph to explain our aims more clearly. We disagree that that the title is misleading as it focuses specifically on climate response scenarios which are the subject of our paper.

—

2) Following the above point, the series of IPCC reports in WG3 has shown the climate implications with probability and uncertain ranges of climate models. In the assessment, climate sensitivity has been already considered to generate the parameter ensemble of simple climate models sampling various parameters simultaneously. Thus, figure 1 panel c has been already addressed. The new thing here would be to show the climate outcomes explicitly comparing with climate sensitivity. Perhaps, it would be new but it needs confirmation from the climate expert.

RESPONSE: Please see our response to the previous point. For the reasons summarised there (and see also Sutton 2018 & 2019), describing climate implications using "probability and uncertain ranges of climate models" is not the most appropriate way to inform risk assessments. Our proposal is that WGI should employ a set of global climate response scenarios (with specific attention to high impact scenarios) and that these same response scenarios should also be investigated and assessed by WGII (to explore consequences for impacts and adaptation) and WGIII (to explore the consequences for mitigation) alongside their use of socio-economic scenarios. We have added this important point at the end of the paper.

—

3) Although it is not the IPCC coordinated activity, it would be worthwhile to acknowledge that in integrated assessment modeling, the discrete scenario proposal has been made and they are now working on that.

RESPONSE: The paper cited is focused on filling gaps in emissions scenarios. It does not address climate response scenarios, which are the focus of our paper.

—

4) Finally, in the paper, I can find the term "each SSPs", but it seems to be a set of SSPs and climate target combination by looking at the figure. For example, SSP1-26 is a combination of SSP1 and radiative forcing target 2.6W.

RESPONSE: We accept this point and have revised the paper and figure to clarify which are the relevant SSPs.

---

## Author Comment (AC2) · 18 May 2020

We thank the referee for their positive comments on our article. We fully agree that the ability of models to simulate historical trends is a critical issue in assessing their credibility and relevance for projections. However, because of the importance of aerosol forcing for past changes, the relationship between historical trends and future warming is not simple. In broad terms, model simulated trends may disagree with observed trends either because the simulated radiative forcing (especially the aerosol component) is incorrect, or because their sensitivity is incorrect, or both. Furthermore, simulated trends may agree well with observed trends but for the wrong reasons (i.e. compensating errors). Consequently, our view is that it is appropriate for IPCC WGI to assess climate sensitivity (ECS and TCR) drawing on multiple lines of evidence in-

cluding historical trends, and then use this assessment to generate climate response scenarios, as in our paper. This procedure does not rely on specific models and would be more robust than past practice such as using the 5-95% CMIP range as the likely range for future projections.

Thank you for pointing out the typo, which we have corrected.

---

## Author Comment (AC3) · 2 Jun 2020

We thank the referee for their thoughtful comments.

Our responses to the specific points about the figure are as follows:

1) In the sample of models used, the lowest ECS is 2.3K so some non-monotonic behaviour in panels d,e is expected, given the lowest climate response scenario considered here is 2.0K.

2) We choose to retain the labels in panels d,e as they are. Changing to labelling the bid edges will not overcome the non-even spacing issue and, in our view, would be more confusing.

[Figure]

3) Panels a,b are necessary to explain the approach used. We have attached for interest a version of the figure which shows GSAT change in 2050 and 2100 relative to pre-industrial, instead of the crossing years in panels d,e. Similar features can be seen: i.e. the same GSAT change can be realised from widely different socio-economic and climate response combinations.

Concerning use of the term scenario - this term is very widely used in problems of risk assessment to describe, and explore the consequences of, a specific set of assumptions about the future. Of course, it is true that in climate science scenarios have traditionally been concerned with socio-economic assumptions only, whereas uncertainties about the climate response have been characterised in terms of likelihood. However, it a fundamental part of our argument that this asymmetric approach is not justified. Future socio-economics and future climate response are both forms of epistemic uncertainty, and it is therefore appropriate to use scenarios for both. To quote Sutton (BAMS, 2019):

"...for the purposes of risk assessment, there is little difference between our knowledge/ ignorance of (say) future population growth and our knowledge/ignorance of (say) the future rate of global warming, so it would be helpful for decision-makers if the same tools - scenarios - were used to communicate this knowledge. Such an approach would be in line with King et al.'s (2015) fifth principle of risk assessment: take a holistic approach. Decision-relevant climate scenarios could usefully be developed to sample all the major dimensions of epistemic uncertainty (e.g., rapid economic growth, high greenhouse gas emissions, and high climate sensitivity)."
* * *
Simple GSAT projections based on ECS
(relative to 1850-1900 baseline)

**(a) 2041-2060**

**(b) 2080-2099**

**(c) Climate response scenarios**

SSP1-1.9: 9 models
SSP1-2.6: 15
SSP2-4.5: 17
SSP3-7.0: 14
SSP5-8.5: 16

ECS

**(d) GSAT 2050**

**(e) GSAT 2090**

SSP1-1.9
SSP1-2.6
SSP2-4.5
SSP3-7.0
SSP5-8.5

**Fig. 1.**

---

## Author Comment (AC4) · 2 Jun 2020

We thank the referee for their positive comments on our article. There is an important issue underlying these comments. Scenario (or "storyline") based approaches and likelihood-based approaches are two alternative methods for describing epistemic uncertainties about the future (e.g. Shepherd et al, 2018). In the former, no quantitative likelihood is attached to specific scenarios – they are merely plausible storylines about how the world might unfold. Nevertheless, they are very useful tools to identify vulnerabilities and risks (e.g. Bank of England "stress tests"), and to assess potential responses.

Traditionally physical climate science has used a scenario-based approach for describ-

ing socio–economic uncertainties and a likelihood-based approach for describing un-certainties in the climate response. We argue that this asymmetric approach is not justified: future socio-economics and future climate response are both forms of epis-temic uncertainty, and it is therefore appropriate to use scenarios for both (see Sutton, BAMS, 2019 for further discussion).

Consequently, it is not the case that we selected five different ECS levels "and repre-sent them with equal weight". We do not weight them at all, and it is not appropriate to do so. Similarly, it is not the case that the different socio-economic scenarios "are not all equally likely" – there is no quantified likelihood associated with each.

Reference: Shepherd, T. G., and Coauthors, 2018: Storylines: An alternative approach to representing uncertainty in physical aspects of climate change. Climatic Change, 151, 555–571, https://doi.org/10.1007/s10584-018 -2317-9.

---

## Author Response (AR1)

**Interactive comments on "ESD Ideas: Global climate response scenarios for IPCC AR6" by Rowan T. Sutton and Ed Hawkins**

**Anonymous Referee #1 Received and published: 4 March 2020**

This article proposes a new methodology to fill the gap of the current existing IPCC scenarios. I should admit that I am not a climate modeler and have expertise in integrated assessment modeling. Thus I can comment from that perspective and need other climate expert's points of view. I list the main arguments below.

1) The current title and abstract lead misunderstanding of the contents and should be reconsidered. In the abstract, there is a problem statement "but the IPCC has not developed similar discrete scenarios". However, the issue is wrongly stated. First, after SRES IPCC never develops scenarios. IAMC (Integrated Assessment Modeling Consortium) generated RCPs and SSPs during the last decade but they are certainly different from IPCC. Second, assuming that the RCPs and SSPs are a set of scenarios addressed by the authors here, the authors are correct within the CMIP context in a way that the socioeconomic scenarios. To this end, a series of past IPCC assessment reports used simple climate models to represent CMIP3 and CMIP5 climate model behavior with ranges of embedded parameter ensembles. So, the issue is neither nonexistence of discrete scenarios nor how to use the scenarios. I think what the authors demonstrate in this paper is just one of the examples of how to use the CMIP results (not the scenario issue).

**We thank the referee for their thoughtful comments on our article.**

We accept that IPCC did not itself generate the RCPs and SSPs and will correct this point. We also agree that simpler climate models have often been used to explore a wider range of socio-economic scenarios than is possible with complex (CMIP-class) climate models, although WGI (which is the focus of our Idea) has placed great emphasis on the specific set of socio-economic scenarios used for CMIP.

However, the focus of our Idea is *not* socio-economic scenarios at all but rather the representation by IPCC of information about uncertainty in the *climate response* to anthropogenic forcing. In past assessment reports this information has usually been expressed in terms of a *likely* range based on CMIP results, e.g. the likely range for global mean surface temperatures under a particular socio-economic scenario. Our argument (following Sutton 2018 & 2019) is that such an approach is flawed because it does not meet policy-maker needs for risk assessments which require specific attention to high impact scenarios even if they are considered *unlikely* to arise (noting that according to IPCC calibrated language unlikely means only

Simple GSAT projections based on ECS (relative to 1850-1900 baseline)

While I understand the motivation and appeal of exploring the joint dependence of a climate impact on two "scenarios" – socio-economic and climate response (or forcing and response), I'm not really a fan of the phrase "climate response scenario" and would prefer that "scenario" be reserved for SSPs. Scenarios in the climate context have historically referred to plausible future social-economic futures (SRES, RCP, SSP) for which humans have some role in determining. ECS is a different beast. Could it just be referred to as "climate response"?

The term scenario is very widely used in problems of risk assessment to describe, and explore the consequences of, a specific set of assumptions about the future. Of course, it is true that in climate science scenarios have traditionally been concerned with socio-economic assumptions only, whereas uncertainties about the climate response have been characterised in terms of likelihood. However, it a fundamental part of our argument that this asymmetric approach is not justified. Future socio-economics and future climate response are *both* forms of epistemic uncertainty, and it is therefore appropriate to use scenarios for both. To quote Sutton (BAMS, 2019):

"...for the purposes of risk assessment, there is little difference between our knowledge/ ignorance of (say) future population growth and our knowledge/ignorance of (say) the future rate of global warming, so it would be helpful for decision-makers if the same tools—scenarios—were used to communicate this knowledge. Such an approach would be in line with King et al.'s (2015) fifth principle of risk assessment: take a holistic approach. Decision-relevant climate scenarios could usefully be developed to sample all the major dimensions of epistemic uncertainty (e.g., rapid economic growth, high greenhouse gas emissions, and high climate sensitivity)."

**Anonymous Referee #4 Received and published: 11 May 2020**

This paper by Sutton and Hawkins presents a new proposal to better reflect geophysical uncertainties that surround the socioeconomic scenario uncertainty in projections of future climate change. The aim of the proposal is to provide more useful information for risk assessments.

The paper in my view can potentially provide a useful contribution to the framing of scenario projections. The suggested visualisation that shows how different thresholds are exceeded as a function of both the forcing and the uncertainty in the response, can provide useful additional insights to users of climate change data. At the same time, the contribution is very succinct and important aspects that are key components of risk assessments are not being covered or discussed. An important aspect in risk assessments is the likelihood connected to a certain hazard. In the proposal presented here, this dimension is not reflected upon. The authors have selected five different ECS levels, and represent them with equal weight. The same is true for the scenarios for which it is obvious that not all are equally likely. This raises immediate additional questions and issues. In particular, the suggested approach creates a great opportunity for biased representation of scientific evidence, which is skewed towards arbitrary selected extremes.

It would therefore be useful if the authors could give this aspect a bit more thought and come up with a suggestion of how likelihood (at least in the geophysical sense) could be incorporated in the visualisations in support of risk assessments

We thank the referee for their positive comments on our article. There is an important issue underlying these comments. Scenario (or "storyline") based approaches and likelihood-based approaches are two alternative methods for describing epistemic uncertainties about the future (e.g. Shepherd et al, 2018, now cited in the revised paper). In the former, no quantitative likelihood is attached to specific scenarios – they are merely plausible storylines about how the world might unfold. Nevertheless, they are very useful tools to identify vulnerabilities and risks (e.g. Bank of England "stress tests"), and to assess potential responses.

Traditionally physical climate science has used a scenario-based approach for describing socioeconomic uncertainties and a likelihood-based approach for describing uncertainties in the climate response. We argue that this asymmetric approach is not justified: future socio-economics and future climate response are *both* forms of epistemic uncertainty, and it is therefore appropriate to use scenarios for both (see Sutton, BAMS, 2019 for further discussion).

Consequently, it is not the case that we selected five different ECS levels "and represent them with equal weight". We do not weight them at all, and it is not appropriate to do so. Similarly, it is not the case that the different socio-economic scenarios "are not all equally likely" – there is no quantified likelihood associated with each.

Reference: Shepherd, T. G., and Coauthors, 2018: Storylines: An alternative approach to representing uncertainty in physical aspects of climate change. *Climatic Change*, **151**, 555–571, https://doi.org/10.1007/s10584-018 -2317-9.

**Summary of changes made to the manuscript**

- 1. Revised abstract, introduction and final paragraph to address comments of all referees, particularly referee 1. Added referenced to Shepherd et al (2018).
- 2. Updated the figure to address comments of referee 1
- 3. Full manuscript with tracked changes follows

[revised manuscript text omitted]

**Simple GSAT projections based on ECS (relative to 1850-1900 baseline)**